# Breed-Specific Variations in Vertebral Right Heart Index (VRHi): Implications for Detection of True and False Right Heart Enlargement (RHE) in Dogs

**DOI:** 10.3390/vetsci12040300

**Published:** 2025-03-24

**Authors:** Kawon Choi, Jaehwan Kim, Kidong Eom, Jongwon Koo, Inseong Jeong, Chul Park

**Affiliations:** 1Department of Veterinary Medical Imaging, College of Veterinary Medicine, Konkuk University, Seoul 05029, Republic of Korea; kawon0122@konkuk.ac.kr (K.C.);; 2Royal Animal Medical Center, Seoul 02140, Republic of Korea

**Keywords:** right heart enlargement, vertebral right heart index, VRHi, brachycephalic, Schnauzer, Dachshund, Pomeranian, Yorkshire Terrier, Miniature Pinscher, thoracic radiography

## Abstract

This study explored breed-specific differences in vertebral right heart index (VRHi) values and their accuracy in diagnosing right heart enlargement (RHE) in dogs. Certain breeds, particularly brachycephalic dogs, often exhibited higher VRHi values despite having a normal heart size. Right lateral thoracic radiographs were more reliable for VRHi measurements than ventrodorsal views. These results emphasize the importance of accounting for breed-specific factors when interpreting chest X-rays to assess heart size in veterinary practice.

## 1. Introduction

In veterinary medicine, the radiographic evaluation of right heart enlargement (RHE) is largely subjective, relying on the assessment of cardiac silhouette margins and shape [1]. In ventrodorsal (VD) or dorsoventral (DV) projections, an enlarged right atrium may appear as an increased bulge along the right heart border between the 9 and 11 o’clock positions. Similarly, a hypertrophic right ventricle often appears more rounded and protrudes into the right hemithorax, creating a reversed D-shaped cardiac silhouette [1].

Based on the authors’ experience, in certain breeds, such as brachycephalic dogs, Schnauzers, Dachshunds, Pomeranians, Yorkshire Terriers, and Miniature Pinschers, radiographic images frequently suggest apparent RHE (e.g., right heart bulging) despite a normal cardiac size and function. The non-brachycephalic group (NBC) refers to breeds other than these specific ones and does not exhibit this pattern. While vertebral heart score (VHS) reference values have been extensively documented for certain breeds [2], false-positive RHE findings in specific breeds have not been systemically reported, nor have any objective measurements for this phenomenon been established.

The vertebral right heart index (VRHi) was recently proposed as a new quantitative radiographic method for the diagnosis of RHE in dogs. However, current studies lack breed-specific or thoracic conformation-based VRHi thresholds, with generalized cutoff values set at right lateral (RL)/left lateral VRHi ≥ 3.5 vertebral units [v] or VD VRHi ≥ 3.0 v [3].

This study aims to

(a)Investigate breed-specific variations in VRHi among dogs with apparent radiographic RHE but a normal cardiac anatomy;(b)Compare the VRHi values between dogs with and without RHE in the NBC and brachycephalic breeds;(c)Establish VRHi cutoff values to differentiate true RHE from false-positive cases;(d)Examine factors contributing to false-positive RHE diagnoses, including echocardiographic parameters and thoracic conformation.

## 2. Materials and Methods

### 2.1. Data Recording Analysis

In this multicenter, retrospective study, client-owned dogs were obtained from three veterinary centers—Konkuk Veterinary Teaching Hospital, the Royal Animal Medical Center, and the Nowon N Animal Medical Center—between 2015 and 2024. The distribution of cases among the centers was as follows: Konkuk Veterinary Teaching Hospital provided 234 cases (65.2%), the Royal Animal Medical Center contributed 87 cases (24.2%), and the Nowon N Animal Medical Center supplied 38 cases (10.6%), for a total of 359 dogs included in the study. The inclusion criteria for dogs without apparent RHE were as follows: unremarkable findings on physical examination, cardiovascular assessment, and echocardiography or a diagnosis of stage B1 myxomatous mitral valve disease without evidence of cardiac remodeling that could affect the right heart appearance on radiographs [4]. Furthermore, individuals presenting with underlying conditions (e.g., coughing, dyspnea, signs of pulmonary hypertension, or right heart remodeling) that could potentially confound the experimental outcomes were excluded from the study cohort. However, patients with trivial or physiological tricuspid regurgitation were included, as these conditions were not indicative of pulmonary hypertension and were deemed unlikely to affect the study results [5]. Data collected for each patient included the breed, sex, age, and body weight. With respect to sample collection, no restrictions were imposed on the age or weight range, and subjects were included irrespective of their neutering status.

All dogs underwent radiographic imaging and echocardiography on the same day. Each patient had RL and VD thoracic radiographic views, and all radiographic views were deemed suitable for measurement. Exclusion criteria were the presence of abnormalities that could obscure the cardiac silhouette, such as pericardial or pleural effusion, malpositioned radiographs (e.g., thoracic rotation along its long axis), or mediastinal or heart base tumors.

### 2.2. Data Collection and Thoracic Radiography Evaluation

In the RL thoracic radiograph, the long axis of the cardiac silhouette was identified as previously described for the VHS [6], and the short axis was measured as the distance from the cranial margin of the cardiac silhouette to the long-axis line. The RL VRHi was calculated as the number of vertebral units equivalent to the short-axis length aligned with thoracic vertebrae starting at the cranial edge of the fourth thoracic vertebra (Figure 1C) [3].

For VD radiographs, the thoracic longitudinal axis was drawn superimposed on the vertebral column, while the maximal transverse axis of the heart was drawn perpendicular to this line. The VD VRHi was determined as the number of thoracic vertebral units in the lateral view, equivalent to the distance from the right margin of the cardiac silhouette to the thoracic longitudinal axis (Figure 1C) [3]. The VRHi measurements were rounded to the nearest 0.1 v.

The thoracic depth-to-width ratio was calculated to determine the thoracic shape and investigate the relationship between chest conformation and the VRHi. The depth was measured on the RL thoracic radiograph as the perpendicular distance from the xiphoid process to the ventral margin of the vertebral body. The width was measured on the VD thoracic radiograph as the distance between the medial borders of the 8th ribs where they met the thoracic wall. Chest conformation was classified based on the depth-to-width ratio: a ratio < 0.75 was categorized as a barrel chest, while a ratio > 1.25 was categorized as a deep chest [7,8,9].

The RL/VD VRHi and thoracic depth-to-width ratios were measured using commercially available image-viewing software, including RadiAnt 2024.2 (Medixant, Poznan, Poland) and INFINITT 7.0 (Infinitt Healthcare, Seoul, Republic of Korea). All radiographic measurements were obtained by a single radiologist (D.V.M.) under the supervision of a senior radiologist with more than 10 years of radiology experience.

### 2.3. Data Collection and Echocardiographic Evaluation

All echocardiographic examinations were conducted on unsedated dogs in right and left lateral recumbency, with simultaneous ECG tracings recorded. Each dog underwent a comprehensive echocardiographic examination, including transthoracic 2D, M-mode, spectral, and color Doppler imaging [10].

The end-diastolic RV area (RVEDA), right atrial area (RAA), and end-diastolic right ventricular wall thickness (RVWTd) were measured as morphological indicators of the right heart size. These measurements were obtained from the left parasternal apical four-chamber view [11,12]. The RVEDA was measured by tracing the endocardial border of the RV inflow region at end-diastole, excluding papillary muscles (Figure 1A) [12,13]. The RAA was measured by tracing the endocardial border of the right atrium at end-systole (Figure 1A) [14]. The RVWTd was measured as the largest diameter of the right ventricular free wall at end-diastole using the B-mode method (Figure 1A) [15]. To account for body size, the RVEDA and RAA indices were normalized using the following formulas:RVEDA index = (RVEDA [cm^2^])/(body weight [kg])^0.624^(1)RAA index = (RAA [cm^2^])/(body weight [kg])^0.714^(2)RVWTd index = (RVWTd [cm])/(body weight [kg])^0.250^
(3)

RHE was defined by right atrial enlargement, right ventricular hypertrophy, and/or enlargement. Right ventricular hypertrophy and/or enlargement was defined as an RVWTd index (cm/kg^0.250^) > 0.39 and/or an RVEDA index (cm^2^/kg^0.624^) > 1.4, while right atrial enlargement was defined as an RAA index (cm^2^/kg^0.714^) > 0.76 [3,11,16].

The RVWTd, RVEDA, and RAA indices were measured using commercially available image-viewing software, including RadiAnt (Medixant) and INFINITT (Infinitt Healthcare, Seoul, Republic of Korea). All echocardiographic measurements were obtained by a single radiologist (D.V.M.) under the supervision of a senior radiologist with more than 10 years of radiology experience.

### 2.4. Statistical Analysis

Continuous variables, including age and body weight, were assessed for a normal distribution using the Shapiro–Wilk test. Numerical data are presented as means with standard deviations or medians with ranges, as appropriate.

Unpaired *t*-tests were used to compare the RL/VD VRHi and echocardiographic parameters (RVWTd, RAA, RVEDA index) between the NBC and other specific breeds (e.g., brachycephalic dogs, Schnauzers, Dachshunds, Pomeranians, Yorkshire Terriers, and Miniature Pinschers). Additionally, *t*-tests were used to compare differences in the RL/VD VRHi between dogs with and without RHE in the NBC and brachycephalic groups.

A receiver operating characteristic (ROC) curve analysis was used to evaluate the diagnostic accuracy of the RL and VD VRHi in detecting RHE. The area under the ROC curve (AUC) served as a summary measure for diagnostic performance. Cutoffs for each variable were determined based on the highest sensitivity and specificity combination using the Youden index.

A linear regression analysis was performed to assess the relationships between body weight and the VRHi and between the depth-to-width ratio and VRHi.

A veterinarian (K.W.C.) performed all statistical analyses using the SPSS statistical software (version 30.0; IBM Corp., Armonk, NY, USA). A *p* value of <0.05 was considered statistically significant.

## 3. Results

### 3.1. Study Population

A total of 359 dogs met the inclusion criteria. The mean age was 10.6 years (range: 1–19 years), and the mean body weight was 6.1 kg (range: 1.2–37.0 kg). The study population included 152 castrated males, 169 spayed females, 19 intact males, and 19 intact females.

The breed groups were classified as follows: brachycephalic breeds, Schnauzers, Dachshunds, Pomeranians, Yorkshire Terriers, Miniature Pinschers, and NBC (which included breeds other than those listed above). Table 1 summarizes each breed group’s demographic and clinical characteristics, including the mean age and weight.

Within the brachycephalic group, there were 118 dogs, including Shih Tzus (66), Chihuahuas (27), Pekingese (13), French Bulldogs (5), Boston Terriers (3), and one Bulldog, King Charles Spaniel, and Pug. The NBC group comprised 87 dogs from various breeds, including Maltese (27), Poodles (18), mixed breeds (14), Cocker Spaniels (5), Spitz (4), Beagles (3), Bichon Frise (3), English Cocker Spaniels (2), Golden Retrievers (2), Silky Terriers (2), and one Airedale Terrier, Border Collie, Jindo, Labrador Retriever, Samoyed, Scottish Terrier, Siberian Husky, and Welsh Corgi. Additionally, the study included Schnauzers (31), Dachshunds (31), Pomeranians (30), Yorkshire Terriers (30), and Miniature Pinschers (32).

In both the brachycephalic and NBC groups, dogs were further classified based on the presence or absence of RHE. Of the 118 brachycephalic dogs, 71 were classified as not having RHE, while 47 were classified as having RHE. In the NBC group, 64 dogs did not exhibit RHE, whereas 23 dogs were diagnosed with RHE. The RHE group comprised 70 dogs with conditions such as pulmonary hypertension (*n* = 61), tricuspid dysplasia (*n* = 8), and pulmonary stenosis (*n* = 1). Pulmonary hypertension (PH) was classified based on the 2020 ACVIM consensus statement into five groups: PH group 1 (*n* = 10), group 2 (*n* = 8), group 3 (*n* = 26), group 4 (*n* = 1), and group 5 (*n* = 16) [5]. Further experimental results are provided in the Appendix A accompanying this article.

Statistical analyses revealed no significant age differences between NBC dogs with RHE and the other groups. No significant differences in sex distribution were observed among the groups. Similarly, no significant differences in body weight were detected between dogs with and without RHE in both brachycephalic and NBC groups.

### 3.2. Imaging and Statistical Analysis

Radiographic studies included RL and VD thoracic radiographs for all 359 dogs. Thoracic depth-to-width ratios were calculated only for dogs without RHE, and those with sternal deformities were excluded. Echocardiographic parameters, including the RAA, RVWTd, and RVEDA indices, were measured for RHE diagnosis.

For dogs without RHE, comparisons between the NBC group and other specific breeds revealed significant differences in the VRHi values. The mean, SD values, and *p*-values for the VRHi of each group are summarized in Table 2. The RL VRHi mean values for the NBC group were significantly lower (3.07 ± 0.24 v) than those for all other breeds, including brachycephalic breeds (3.29 ± 0.33 v), Schnauzers (3.55 ± 0.32 v), Dachshunds (3.34 ± 0.33 v), Pomeranians (3.48 ± 0.33 v), Yorkshire Terriers (3.32 ± 0.35 v), and Miniature Pinschers (3.48 ± 0.34 v) (all *p* < 0.001) (Figure 1 and Figure 2). For the VD VRHi, the mean values for the NBC group (2.32 ± 0.36 v) were significantly lower than those of Schnauzers, Pomeranians, Yorkshire Terriers, and Miniature Pinschers (all *p* < 0.001), but not significantly different from those of brachycephalic breeds (2.43 ± 0.43 v) or Dachshunds (2.34 ± 0.37 v).

However, Schnauzers, Pomeranians, Yorkshire Terriers, and Miniature Pinschers demonstrated significantly higher VD VRHi values compared to those of the NBC group (mean ± SD: 2.80 ± 0.40 v, 2.71 ± 0.42 v, 2.67 ± 0.35 v, and 2.66 ± 0.34 v, respectively; all *p* < 0.001). These differences are illustrated in Figure 2 and Figure 3. In contrast, there were no significant differences in the echocardiographic indices (RAA, RVWTd, RVEDA index) when comparing the NBC group to all other breed groups. The mean and SD values for the echocardiographic indices (RAA, RVWTd, RVEDA index) of each group are summarized in Table 3.

The linear regression analysis revealed no significant relationship between body weight and VRHi measurements for the RL VRHi (R = 0.072; *p* = 0.21) and VD VRHi (R = 0.038; *p* = 0.51). Similarly, the depth-to-width ratio showed very weak, negative correlations with the RL VRHi (R = −0.147; *p* = 0.01) and VD VRHi (R = −0.194; *p* < 0.05). While the RL VRHi did not differ significantly between dogs with and without barrel chests (*p* = 0.217), the VD VRHi showed a statistically significant difference between the two groups (*p* = 0.045). Within the NBC group, no significant association was observed between the presence or absence of barrel chest and the VRHi measurements for either the RL VRHi (*p* = 0.057) or VD VRHi (*p* = 0.552).

In the brachycephalic and NBC groups, individuals with RHE exhibited significantly higher VRHi values compared to those without RHE (all *p* < 0.001) (Figure 4). These findings are summarized in Table 4 and visualized in Figure 4.

The optimal cutoffs of the RL VRHi and VD VRHi to detect RHE are summarized in Table 5. The ROC curve analysis, conducted separately for the brachycephalic and NBC groups, revealed minimal differences in the AUC for the RL and VD VRHi in brachycephalic breeds (AUC = 0.80 vs. 0.81) (Figure 5B). In contrast, for the NBC group, the RL VRHi showed significantly higher diagnostic accuracy (AUC = 0.9) compared to the VD VRHi (AUC = 0.7) (Figure 5A).

## 4. Discussion

Based on the authors’ experience, certain breeds, including brachycephalic breeds, Schnauzers, Dachshunds, Pomeranians, Yorkshire Terriers, and Miniature Pinschers, frequently display radiographic signs of apparent RHE despite having a normal cardiac size and function. To the best of our knowledge, this phenomenon of false RHE in specific breeds has not been previously documented in the veterinary literature, nor have objective measurements been reported to address this discrepancy. While the diagnostic criteria for left-sided heart enlargement have been extensively studied and validated [6,17,18,19,20,21], the absence of established quantitative radiological standards for RHE has likely contributed to the variability and subjectivity in interpreting these findings [22]. The recently proposed VRHi offers a quantitative radiographic method for the assessment of RHE in dogs, marking a significant advancement in addressing this diagnostic gap [3].

This study demonstrated that specific breeds, including brachycephalic breeds, Schnauzers, Dachshunds, Pomeranians, Yorkshire Terriers, and Miniature Pinschers, displayed significantly elevated VRHi values compared to those of the NBC group, despite the absence of echocardiographic evidence of RHE. The RL VRHi was consistently higher across all these breeds than in the NBC group. At the same time, the VD VRHi showed significantly increased values in Schnauzers, Pomeranians, Yorkshire Terriers, and Miniature Pinschers compared to those from the NBC group. In contrast, echocardiographic parameters, including the RAA, RVEDA, and RVWTd indices, showed no statistically significant differences between the NBC group and the other breeds. These findings highlight that radiographic signs of RHE may not correspond to true anatomical enlargement of the right heart chambers.

Several hypotheses may explain these findings. Brachycephalic dogs and barrel-chested breeds are known to have shorter vertebral bodies, which may result in elevated VRHi values due to the overestimation of the cardiac size when using vertebral-based measurements [8]. However, this study’s depth-to-width ratio showed no consistent linear relationship with the VRHi values, suggesting that factors beyond the vertebral body length may have influenced the results. One such factor could be the heart’s vertical axis rotation, which can significantly alter the radiographic appearance of cardiac structures and mimic RHE even in its absence [23,24]. Notably, Schnauzers demonstrated the highest mean RL VRHi value (3.55 v) among breeds with normal hearts, supporting the hypothesis that vertical axis rotation, including a leftward deviation in the cardiac apex, may contribute to the elevated VRHi values in this breed (Figure 1C). Furthermore, the statistically significant association between barrel chest and the VD VRHi suggests that the cardiac axis in barrel-chested breeds may affect the VD radiographic perspective. These anatomical variations underscore the need for further studies to confirm these hypotheses and clarify the mechanisms underlying breed-specific VRHi elevations.

The comparative analysis of dogs with and without RHE revealed statistically significant differences in the RL and VD VRHi across the brachycephalic and NBC cohorts. The ROC curve analysis demonstrated no significant difference in the AUC for the RL or VD VRHi in brachycephalic breeds. However, in NBC breeds, the RL VRHi demonstrated superior diagnostic accuracy compared to the VD VRHi. Moreover, the RL VRHi showed a stronger correlation with the echocardiographic indices than the VD VRHi, suggesting that the RL VRHi is a key parameter for the accurate radiographic assessment of the cardiac size, particularly in NBC breeds.

While previous studies have quantified radiographic indices [3], they did not stratify the VRHi values by breed, chest morphology, or skull type. Moreover, the discrepancies in the cutoff values between these studies and our findings are likely attributable to the breed-specific categorization employed in this study. Notably, Schnauzers exhibited a mean RL VRHi value of 3.55 v for normal hearts, which exceeds the previously published reference cutoff (3.5 v). This highlights the importance of accounting for breed-specific variations in radiographic cardiac assessment. Although traditional diagnostic approaches often rely on identifying radiographic signs such as cardiac bulging, certain breeds may present these features despite normal echocardiographic findings. Despite these limitations, radiography remains the primary imaging modality for the assessment of RHE in clinical veterinary practice. This study demonstrated that, in breeds predisposed to exhibiting false radiographic RHE, individuals may exhibit VRHi values exceeding the published reference ranges yet show normal echocardiographic findings, particularly in the absence of cardiac murmurs, respiratory symptoms, or left-sided heart failure. These results emphasize the need for a comprehensive clinical assessment to complement radiographic interpretation, especially in such predisposed breeds. In clinical practice, these findings highlight the importance of integrating echocardiographic evaluation for breeds prone to false RHE on radiographs. This approach may help to avoid over-diagnosing cardiac conditions and initiating unnecessary treatments, particularly in asymptomatic animals without murmurs or respiratory symptoms. Additionally, the establishment of breed-specific VRHi cutoff values can enhance the diagnostic accuracy, reducing the need for unnecessary referrals or advanced imaging in cases where the radiographic findings are ambiguous.

This study has several limitations. First, the criterion for the determination of RHE relied on echocardiographic assessment using the left apical four-chamber view, instead of the RV-focused view commonly utilized in evaluating the RV. This choice aligns with standard veterinary echocardiographic practice, as the left apical four-chamber view is widely used for its practicality and comprehensive visualization of the heart chambers, although it may not capture the complexity of the right ventricular geometry as effectively as the RV-focused view. Research in human medicine indicates that the left apical four-chamber view underestimates measurements compared to the RV-focused view [25]. However, this methodological choice likely reduced the risk of falsely diagnosing RHE when it was not present. Second, due to limited sample sizes, the study could not establish cutoff values for Schnauzers, Dachshunds, Pomeranians, Yorkshire Terriers, and Miniature Pinschers. Larger studies focusing on these breeds with RHE are needed to derive more precise diagnostic criteria. Third, an interobserver reliability study was not conducted for the 359 dogs included, which could limit the generalizability of these findings. Future research should incorporate interobserver variability analyses to enhance the robustness and reproducibility of the results. Moreover, the NBC group included various breeds (mesocephalic, dolichocephalic, and mixed), which could potentially have influenced the results. We examined the statistical significance of the VRHi values within the NBC group based on barrel chest presence, but found no significant differences. Further research is needed to investigate the relationship between the cephalic index and VRHi across different breed morphologies [26]. Finally, while this study evaluated 30 breeds, further investigation is required to determine whether other breeds may exhibit apparent radiographic RHE without actual enlargement or anatomical changes. Such studies would contribute to a broader understanding of breed-specific radiographic variations in cardiac morphology.

## 5. Conclusions

Certain breeds, such as brachycephalic types, Schnauzers, Dachshunds, Pomeranians, Yorkshire Terriers, and Miniature Pinschers, often show elevated VRHi values without actual right heart enlargement. This study emphasizes the need for breed-specific radiographic standards to improve the diagnostic accuracy. Future studies should aim to establish breed-specific cutoff values and investigate the anatomical factors contributing to these radiographic variations.

## Figures and Tables

**Figure 1 vetsci-12-00300-f001:**
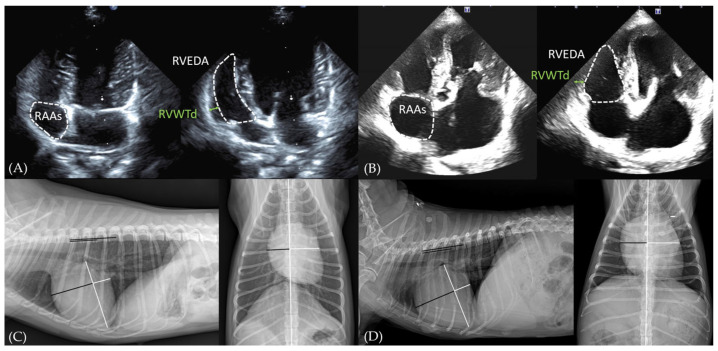
Echocardiographic (**A**,**B**) and thoracic radiographic (**C**,**D**) images showing the methods used to measure the right ventricular wall thickness, right ventricular area, and right atrial area in the study. The left image (**A**) illustrates the measurement of the right atrial area (white dotted line) from the left apical long-axis view in end-systole, and the right image (**B**) shows the measurement of the right ventricular area (white dotted line) and right ventricular wall thickness (green double-headed arrow) in end-diastole from the same view. (**C**) represents a brachycephalic dog with normal echocardiographic parameters but abnormal VRHi values (RL VRHi: 3.3 v, VD WRHi: 2.5 v; black line). (**D**) shows a dog with echocardiographic right heart enlargement and abnormal VRHi values (RL VRHi: 4.1 v, VD VRHi: 2.7 v; black line).

**Figure 2 vetsci-12-00300-f002:**
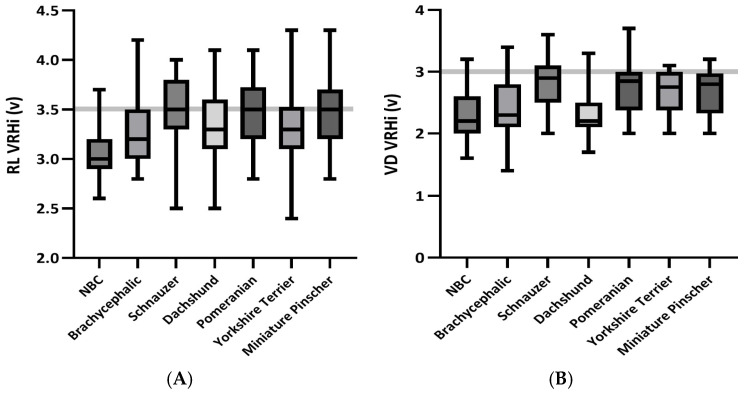
Box plots representing vertebral right heart index (VRHi) measurements for each breed group without right heart enlargement (RHE). (**A**) Right lateral (RL) VRHi and (**B**) ventrodorsal (VD) VRHi measurements are shown for each group. The solid black line within each box plot represents the median VRHi for the respective breed. The horizontal dashed grey line indicates the published reference cutoff values for RL VRHi (3.5 v) and VD VRHi (3.0 v). Abbreviations: NBC, non-brachycephalic group for breeds not belonging to the following groups: Schnauzer, Dachshund, Pomeranian, Yorkshire Terrier, and Miniature Pinscher; v, vertebral units.

**Figure 3 vetsci-12-00300-f003:**
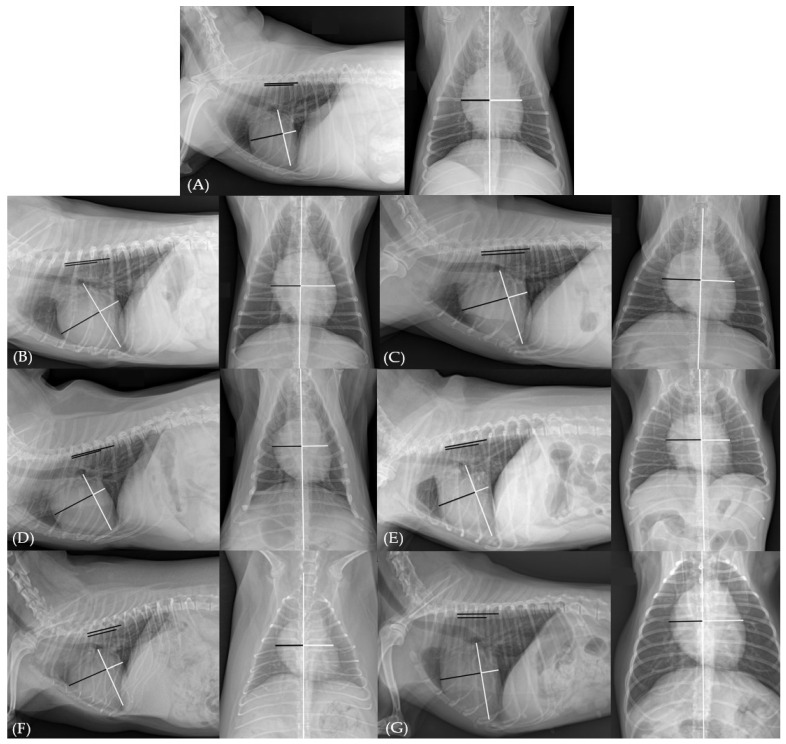
Representative thoracic radiographs illustrating right lateral (RL) and ventrodorsal (VD) vertebral right heart index (VRHi) (black lines) across different breed groups. (**A**) Non-brachycephalic (NBC) dog without right heart enlargement (RHE) showing normal VRHi values (RL VRHi: 3.3 vertebral units [v], VD VRHi: 2.9 v). (**B**) Brachycephalic dog with borderline RL VRHi (3.5 v) and normal VD VRHi (2.5 v). (**C**) Schnauzer exhibiting elevated VRHi in both views (RL VRHi: 4.0 v, VD VRHi 3.5 v). (**D**) Dachshund with elevated RL VRHi (3.7 v) and normal VD VRHi (2.5 v). (**E**) Pomeranian showing borderline RL VRHi (3.6 v) and normal VD VRHi (3.0 v). (**F**) Yorkshire Terrier with borderline VRHi in both views (RL VRHi 3.3 v, VD VRHi 3.0 v). (**G**) Miniature Pinscher demonstrating elevated RL VRHi (3.8 v) and borderline VD VRHi (3.0 v). (Note: Published reference cutoff values for RL and VD VRHi are 3.5 v and 3.0 v, respectively).

**Figure 4 vetsci-12-00300-f004:**
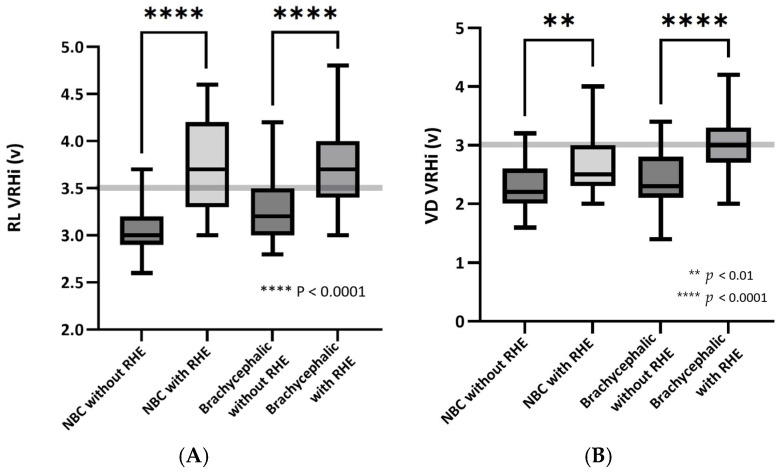
Box-and-whisker plots of vertebral right heart index (VRHi) in brachycephalic and non-brachycephalic (NBC) dogs with and without right heart enlargement (RHE). (**A**) Right lateral (RL) VRHi and (**B**) ventrodorsal (VD) VRHi measurements for brachycephalic and NBC dogs are shown. The solid black line within each box plot indicates the median VRHi for each subgroup. The dashed grey line represents the published reference cutoff values: RL VRHi (3.5 v) and VD VRHi (3.0 v). Statistical significance was determined using unpaired *t*-tests, with significant differences between groups marked as follows: *p* < 0.01 (double asterisks) and *p* < 0.0001 (four asterisks). Abbreviations: NBC, non-brachycephalic group for breeds not belonging to the following groups: Schnauzer, Dachshund, Pomeranian, Yorkshire Terrier, and Miniature Pinscher.

**Figure 5 vetsci-12-00300-f005:**
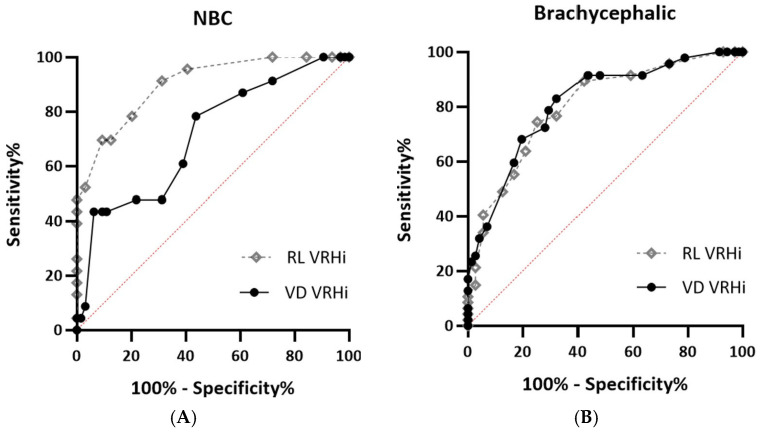
Receiver operating characteristic (ROC) curves for right lateral (RL) and ventrodorsal (VD) vertebral right heart index (VRHi) in the detection of right heart enlargement (RHE). The red dotted line represents the performance of a random classifier, where sensitivity equals 1-specificity. (**A**) ROC curves for the non-brachycephalic group (NBC), excluding Schnauzer, Dachshund, Pomeranian, Yorkshire Terrier, and Miniature Pinscher breeds. RL VRHi demonstrates higher diagnostic accuracy (AUC = 0.90) than VD VRHi (AUC = 0.71). (**B**) ROC curves for brachycephalic breeds, showing similar diagnostic accuracy for RL VRHi (AUC = 0.80) and VD VRHi (AUC = 0.81). The diagonal dotted line represents the reference line for random chance. Abbreviations: AUC: area under the curve; RL: right lateral; VD: ventrodorsal; NBC: NBC, non-brachycephalic group for breeds not belonging to the following groups: Schnauzer, Dachshund, Pomeranian, Yorkshire Terrier, and Miniature Pinscher.

**Table 1 vetsci-12-00300-t001:** Demographic characteristics (age, weight, and sex distribution) of breed groups with and without right heart enlargement (RHE).

Group	M/F (N)	Age (Years) (Mean ± SD, Range)	Weight (kg) (Mean ± SD)
Brachycephalic breeds			
Without RHE	32/39	10.5 ± 3.7 (1–19)	5.6 ± 2.3
With RHE	23/24	11.4 ± 3.5 (1–17)	5.4 ± 2.7
NBC breeds			
Without RHE	34/30	10.3 ± 2.9 (1–14)	7.3 ± 6.2
With RHE	14/9	7.0 ± 3.8 (1–15)	10.3 ± 7.4
Schnauzers	19/12	12.0 ± 2.5 (8–18)	7.4 ± 2.4
Dachshunds	18/13	10.5 ± 3.7 (2–18)	7.4 ± 2.2
Pomeranians	8/22	10.5 ± 3.4 (5–18)	3.5 ± 1.2
Yorkshire Terriers	10/20	11.6 ± 3.0 (3–17)	3.5 ± 1.7
Miniature Pinschers	13/19	11.0 ± 2.5 (6–15)	5.1 ± 1.3

Abbreviations: M, male; F, female; N, number of dogs in each gender; NBC, non-brachycephalic group for breeds not belonging to the following groups: Schnauzer, Dachshund, Pomeranian, Yorkshire Terrier, and Miniature Pinscher; RHE, right heart enlargement.

**Table 2 vetsci-12-00300-t002:** Statistical parameters of vertebral right heart index (VRHi) for each breed group without right heart enlargement (RHE) and comparisons of breed versus NBC mean VRHi.

Group	RL VRHi (Mean ± SD, v)	VD VRHi (Mean ± SD, v)	*p* Value (Breed vs. NBC) *	
			RL VRHi	VD VRHi
NBC	3.07 ± 0.24	2.32 ± 0.36	-	-
Brachycephalic breeds	3.29 ± 0.33	2.43 ± 0.43	<0.001	0.133
Schnauzer	3.55 ± 0.32	2.80 ± 0.40	<0.001	<0.001
Dachshund	3.34 ± 0.33	2.34 ± 0.37	<0.001	0.802
Pomeranian	3.48 ± 0.33	2.71 ± 0.42	<0.001	<0.001
Yorkshire Terrier	3.32 ± 0.35	2.67 ± 0.35	<0.001	<0.001
Miniature Pinscher	3.48 ± 0.34	2.66 ± 0.34	<0.001	<0.001

* *p* < 0.05 is considered statistically significant. Note. “-” indicates that the parameter was not determined. Abbreviations: NBC, non-brachycephalic group for breeds not belonging to the following groups: Schnauzer, Dachshund, Pomeranian, Yorkshire Terrier, and Miniature Pinscher; RHE, right heart enlargement; v, vertebral units.

**Table 3 vetsci-12-00300-t003:** Statistical parameters of echocardiographic indices (RAA, RVWTd, RVEDA index) for each breed group without right heart enlargement (RHE) and comparisons of breed versus NBC.

Group	RVWTd Index (Mean ± SD) (cm/kg^0.250^) (N)	RVEDA Index (Mean ± SD) (cm^2^/kg^0.624^) (N)	RAA Index (Mean ± SD) (cm^2^/kg^0.714^) (N)	*p* Value (Breed vs. NBC) *		
				RVWTd Index	RVEDA Index	RAA Index
NBC	0.301 ± 0.059 (57)	0.593 ± 0.182 (46)	0.433 ± 0.130 (63)	-	-	-
Brachycephalic breeds	0.313 ± 0.040 (67)	0.539 ± 0.202 (54)	0.430 ± 0.150 (69)	0.205	0.164	0.894
Schnauzer	0.288 ± 0.053 (29)	0.618 ± 0.164 (28)	0.441 ± 0.136 (31)	0.314	0.564	0.803
Dachshund	0.280 ± 0.055 (27)	0.628 ± 0.178 (26)	0.400 ± 0.101 (31)	0.122	0.439	0.210
Pomeranian	0.283 ± 0.043 (26)	0.590 ± 0.149 (25)	0.443 ± 0.125 (30)	0.151	0.937	0.744
Yorkshire Terrier	0.295 ± 0.042 (24)	0.616 ± 0.098 (22)	0.462 ± 0.096 (29)	0.641	0.583	0.299
Miniature Pinscher	0.284 ± 0.055 (22)	0.604 ± 0.140 (22)	0.435 ± 0.119 (32)	0.237	0.812	0.943

* *p* < 0.05 is considered statistically significant. Note. “-” indicates that the parameter was not determined. Abbreviations: N, number of dogs in each parameter; NBC, non-brachycephalic group for breeds not belonging to the following groups: Schnauzer, Dachshund, Pomeranian, Yorkshire Terrier, and Miniature Pinscher; RHE, right heart enlargement.

**Table 4 vetsci-12-00300-t004:** Statistical parameters of vertebral right heart index (VRHi) and echocardiographic indices (RAA, RVWTd, RVEDA index) for NBC and brachycephalic breeds with or without RHE and comparisons of the VRHi values of dogs with and without RHE.

Group		RL VRHi (Mean ± SD) (v)	VD VRHi (Mean ± SD) (v)	*p* Value *		RVWTd Index (Mean ± SD) (cm/kg^0.250^) (N)	RVEDA Index (Mean ± SD) (cm^2^/kg^0.624^) (N)	RAA Index (Mean ± SD) (cm^2^/kg^0.714^) (N)
				RL VRHi	VD VRHi			
NBC	Without RHE	3.07 ± 0.24	2.32 ± 0.36	<0.001	<0.001	0.301 ± 0.059 (57)	0.593 ± 0.182 (46)	0.433 ± 0.130 (63)
	With RHE	3.77 ± 0.50	2.65 ± 0.47			0.348 ± 0.053 (23)	1.292 ± 0.340 (23)	1.140 ± 0.485 (23)
Brachycephalic	Without RHE	3.29 ± 0.33	2.43 ± 0.43	<0.001	<0.001	0.313 ± 0.040 (67)	0.539 ± 0.202 (54)	0.430 ± 0.150 (69)
	With RHE	3.74 ± 0.42	3.0 ± 0.46			0.354 ± 0.067 (47)	1.163 ± 0.343 (47)	1.115 ± 0.367 (47)

* *p* < 0.05 is considered statistically significant. Abbreviations: N, number of dogs in each parameter; NBC, non-brachycephalic group for breeds not belonging to the following groups: Schnauzer, Dachshund, Pomeranian, Yorkshire Terrier, and Miniature Pinscher; RHE, right heart enlargement; v, vertebral units.

**Table 5 vetsci-12-00300-t005:** Optimal cutoffs for right lateral and ventrodorsal vertebral right heart index (VRHi) when detecting right heart enlargement within brachycephalic and non-brachycephalic (NBC) groups.

Group		AUC	95% CI	Cutoff (v)	Sn (%)	Sp (%)
Brachycephalic	RL VRHi	0.80	0.72–0.88	≥3.45	75	75
				≥2.95	100	7
				≥4.25	11	100
	VD VRHi	0.81	0.74–0.89	≥2.75	72	72
				≥1.95	100	8
				≥3.45	17	100
NBC	RL VRHi	0.90	0.83–0.97	≥3.25	78	80
				≥2.95	100	28
				≥3.80	48	100
	VD VRHi	0.71	0.59–0.84	≥2.4	61	61
				≥1.95	100	9
				≥3.6	4	100

Abbreviations: AUC, area under the curve; CI, confidence interval; Sn, sensitivity; Sp, specificity; v, vertebral units; RL, right lateral; VD, ventrodorsal; VRHi, vertebral right heart index. NBC, non-brachycephalic group for breeds not belonging to the following groups: Schnauzer, Dachshund, Pomeranian, Yorkshire Terrier, and Miniature Pinscher.

## Data Availability

The data from the present study are available from the corresponding author upon reasonable request.

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
