# Peer review of "Breed-Specific Variations in Vertebral Right Heart Index (VRHi): Implications for Detection of True and False Right Heart Enlargement (RHE) in Dogs"

_vetsci, 2025, doi:10.3390/vetsci12040300_

Round 1
Reviewer 1 Report
Comments and Suggestions for Authors
In the manuscript, the authors' main work is to study multi-breed dogs, and found that the VRHi values of specific breeds such as brachycephalic breeds and schnauzers were significantly higher than those in the non-brachycephalic breed group (NBC), but there was no significant difference in echocardiographic parameters, indicating that RHE image signs do not necessarily correspond to right heart enlargement; Comparing dogs with and without RHE, there were significant differences in RL and VD VRHi and echocardiographic parameters between the brachycephalic and NBC groups, and the diagnostic accuracy of RL VRHi was higher in NBC breeds. The authors' primary contribution is to consider the importance of species-specific variants in radiological cardiac evaluation, to emphasize the significance of comprehensive clinical evaluation and echocardiographic evaluation for false-positive RHE cultivars, and to establish species-specific VRHi cut-offs.
However, the manuscript also has some shortcomings
1. There are two kinds of spelling of the right heart index of the vertebral body in the research objective: "VRHI" and "VRHi", and according to the article.The spelling should be "VRHi". The same problem is shared elsewhere in the article.
2. The study sample involved multi-center data collection, but the number and proportion of samples provided by each center were not mentioned.The distribution of samples from different centers may affect the results. In addition, the non-brachycephalic breed group (NBC) contains many different varieties.Whether their internal differences will affect the results of the study is recommended to be further subdivided or the grouping criteria should be clarified.
3. The selection of study samples did not indicate how the 359 dogs were selected from the overall dog population, whether the specific range of weight and age was restricted, whether they were neutered, and whether there were other underlying diseases, which may have an impact on the experimental results. It is recommended to discuss the possible impact of these potential confounders on the results and whether appropriate controls have been put in place in the study design or analysis.
4. The results mainly present the data in text and tables.It is recommended to add some charts, such as pie charts, to visually show the proportional relationship between each part in the population.
5. The relationship between thoracic cage morphology and VRHi values, it is recommended to supplement anatomical or imaging studies for support.
Author Response
Comment 1: There are two kinds of spelling of the right heart index of the vertebral body in the research objective: "VRHI" and "VRHi", and according to the article.The spelling should be "VRHi". The same problem is shared elsewhere in the article.
Response 1: We sincerely appreciate your thorough review and the valuable feedback provided. We are grateful for bringing this inconsistency to our attention.
In response to your comment regarding the spelling of the right heart index of the vertebral body: We acknowledge the inconsistency in the spelling of "VRHI" and "VRHi" throughout the manuscript. We fully agree that the correct spelling should be "VRHi" as per the convention used in the article. We have carefully reviewed the entire manuscript and made the necessary corrections to ensure consistency.
Specifically, we have made the following changes:
- Line 55: Changed "VRHI" to "VRHi"
- Line 59: Changed "VRHI" to "VRHi"
We have also thoroughly checked the rest of the document to ensure that all instances now correctly read "VRHi". We sincerely apologize for this oversight and thank you for bringing it to our attention. We believe these changes improve the clarity and consistency of our manuscript.
Thank you once again for your valuable input and the opportunity to improve our work. We hope that these revisions meet your expectations.
Comment 2: The study sample involved multi-center data collection, but the number and proportion of samples provided by each center were not mentioned. The distribution of samples from different centers may affect the results. In addition, the non-brachycephalic breed group (NBC) contains many different varieties. Whether their internal differences will affect the results of the study is recommended to be further subdivided or the grouping criteria should be clarified.
Response 2: Thank you for your insightful feedback. I am truly grateful for the valuable points you have raised, which I had not previously considered. Your guidance has been enlightening, and I sincerely appreciate your thoughtful review.
In response to your comments, I have made the following revisions to the manuscript:
- I have incorporated the sample sizes and proportions from the multicenter study as you suggested: "The distribution of cases among the centers was as follows: Konkuk Veterinary Teaching Hospital provided 234 cases (65.2%), Royal Animal Medical Center contributed 87 cases (24.2%), and Nowon N Animal Medical Center supplied 38 cases (10.6%), for a total of 359 dogs included in the study."
- We have conducted additional analysis on the NBC group to investigate the potential influence of barrel chest on VRHi values: "Within the NBC group, no significant association was observed between the presence or absence of barrel chest and VRHi measurements for either RL VRHi (p = 0.057) or VD VRHi (p = 0.552)."
- While we initially separated breeds that might appear to have larger right hearts (e.g., Schnauzers, Dachshunds) from the NBC group, I now realize that we could further refine our categorization to include mesocephalic and dolichocephalic breeds. I have also added a note acknowledging the limitations of the NBC group and the need for further research: "Moreover, the NBC group included various breeds (mesocephalic, dolichocephalic, and mixed), which could potentially influence the results. We examined the statistical significance of VRHi values within the NBC group based on barrel chest presence, but found no significant differences. Further research is needed to investigate the relationship between cephalic index and VRHi across different breed morphologies ."
Thank you once again for your valuable feedback, which has undoubtedly improved the quality of our manuscript. I am grateful for your time and expertise in guiding this research.
Comment 3: The selection of study samples did not indicate how the 359 dogs were selected from the overall dog population, whether the specific range of weight and age was restricted, whether they were neutered, and whether there were other underlying diseases, which may have an impact on the experimental results. It is recommended to discuss the possible impact of these potential confounders on the results and whether appropriate controls have been put in place in the study design or analysis.
Response 3: Thank you for your valuable feedback. I sincerely appreciate your attention to detail and the opportunity to clarify these important points.
In response to your comments, I have made the following additions to the manuscript:
"Furthermore, individuals presenting with underlying conditions (e.g., coughing, dyspnea, signs of pulmonary hypertension such as tricuspid regurgitation, or right heart remodeling) that could potentially confound the experimental outcomes were excluded from the study cohort."
"With respect to sample collection, no restrictions were imposed on age or weight range, and subjects were included irrespective of their neutering status. "
These additions provide greater clarity regarding our exclusion criteria and sample selection process. I believe these changes address your concerns and enhance the overall quality of our manuscript.
Thank you once again for your insightful comments. Your expertise has been invaluable in improving our research presentation.
Comment 4: The results mainly present the data in text and tables. It is recommended to add some charts, such as pie charts, to visually show the proportional relationship between each part in the population.
Response 4: Thank you for your valuable input. I agree that providing the numbers and percentages for each group will indeed be informative for our readers.
In response to your suggestion, we have created a pie chart and added it to the supplementary material. This chart now illustrates:
- The number and percentage of each group
- The number and percentage of breeds within the brachycephalic group
- The number and percentage of breeds within the NBC (non-brachycephalic) group

We believe this addition enhances the understanding of our sample distribution and allows for better interpretation of our data.
Thank you again for your guidance, which has significantly improved the clarity of our manuscript.
Comment 5: The relationship between thoracic cage morphology and VRHi values, it is recommended to supplement anatomical or imaging studies for support.
Response 5: Thank you for your insightful comments. I have given considerable thought to the points you raised, particularly regarding the higher VRHi values observed in brachycephalic and other breeds.
In response to your feedback, I would like to share the following:
- I proposed several hypotheses to explain the elevated VRHi in these breeds, with chest morphology being one of the key factors considered.
Our method for measuring the ratio is as follows:
The thoracic depth-to-width ratio was calculated to determine thoracic shape and investigate the relationship between chest conformation and VRHi [4]. Depth was measured on the RL thoracic radiograph as the perpendicular distance from the xiphoid process to the ventral margin of the vertebral body. The width was measured on the VD thoracic radiograph as the distance between the medial borders of the 8th ribs where they meet the thoracic wall. Chest conformation was classified based on the depth-to-width ratio: a ratio < 0.75 was categorized as a barrel chest, while a ratio > 1.25 was categorized as a deep chest [7–9]
We have also included the literature we referenced for this methodology:
- Carlsson, C.; Häggström, J.; Eriksson, A.; Järvinen, A.-K.; Kvart, C.; Lord, P. Size and Shape of Right Heart Chambers in Mitral Valve Regurgitation in Small-Breed Dogs. J. Vet. Intern. Med. 2009, 23, 1013-1020. https://doi.org/10.1111/j.1939-1676.2009.0359.x
- Corcoran, B.M. Static respiratory compliance in normal dogs. J. Small Anim. Pract. 1991, 32, 438–442.
- Asorey, I.; Pellegrini, L.; Canfrán, S.; Ortiz-Díez, G.; Aguado, D. Factors affecting respiratory system compliance in anaesthetised mechanically ventilated healthy dogs: A retrospective study. J. Small Anim. Pract. 2020, 61, 617–623. https://doi.org/10.1111/jsap.13194.
- Phansangiemjit, A.; Kasemjiwat, K.; Patchanee, K.; Panninvong, Y.; Sunisarud, A.; Choisunirachon, N.; Thanaboonnipat, C. The Differences in Radiographic Vertebral Size in Dogs with Different Chest and Skull Types. Animals 2024, 14, 470. https://doi.org/10.3390/ani14030470
2. To address this, we investigated whether thoracic shape might influence the results. We examined both linear correlations and statistically significant relationships between chest morphology and VRHi. However, our analysis did not reveal any strong significant associations.
3. Following this, we considered cardiac axis as another potential factor. We have acknowledged this as a limitation of our study and highlighted the need for further research in this area.
4. Your comments have prompted an additional consideration: the potential use of CT scans for 3D reconstruction to further investigate these relationships. This could indeed be a valuable approach for future studies.
I appreciate your thought-provoking feedback, which has not only enhanced our current study but also opened avenues for future research. Your expertise has been invaluable in refining our understanding and approach to this complex topic.
Thank you again for your time and dedication to improving our research.
Reviewer 2 Report
Comments and Suggestions for Authors
From my personal opinion, this is an interesting and important study, to remind the reader not to over-interpret VRHi for detecting RHE in dogs, especially in certain breeds.
However, there are some points that could benefit from modification.
And most importantly, I’m confused by the author’s criteria for RHE and non-RHE dogs.
Introduction
Line 43: please list the reference for the aforementioned breeds being more frequently suggested RHE on radiography, especially for those non-brachycephalic breeds.
Material and method
Line70: I believe a more detailed description of the criteria of without signs of pulmonary hypertension should be described more detaily.
What do you mean: without signs of pulmonary hypertension?
For instance: are patients with trivial/physiological TR allowed to be included?, Are patients with trivial Pulmonary insufficiency allowed to be included, Are patients with mild TR but a normal estimated PAP allowed to be included?
I believe more detailed criteria would be beneficial for this manuscript.
Line 71: The author cited the 2024 JVIM study for the right heart remodeling criteria, while this is a retrospective study between 2015 and 2024.
Does the author go back and review all the images and measure the right heart ratio based on the 2024 JVIM study?
If so, how many authors participate in the reviewing process?
I believe a more detailed description would benefit the reader to better understand the criteria.
Line 116, I suggest changing the description to:
right ventricular hypertrophy and/or enlargement was defined as an RVWTd index > 0.39 and/or an RVEDA index > 1.4.
I also suggested the author should clarify how many veterinarians/Authors have participated in the measurement of radiological and echocardiographic parameters, respectively. Multiple observers/raters would affect the variability and replicability of the study, especially when inter-observer variability assessment was not performed.
Results
Line 183 I’m confused about the criteria of RHE and non-RHE groups of the study. In Line 71, the authors cite the 2024 JVIM study as the reference for right heart remodeling, while in Line 116, the authors state the criteria for RHE is based on RVEDA index, RVWTd index, and RAA index. But here the authors state the RAA, RVWTd, and RVEDA could be excluded if image quality was impossible to measure.
So my question is, what exactly are the criteria for the non-RHE group, is it based on the absence of TR and normal right heart ratio, or, based on RVEDA, RVWTd, and RAA.
A detailed description of the criteria of the non-RHE group is needed.
Line 197-198, the author stated there were no significant differences in RAA, RVWTd, and RVEDA when comparing the NBC group to other breeds. I suggested a table with the mean +/- SD of these three parameters would be beneficial for the reader to understand if it is still possible to have a difference in echocardiographic parameters between groups, despite not being statistically significant.
Line 197-198, I also recommend the table should include the number of dogs that could be successfully obtain RAA, RVWTd, and RVEDA values of each group, as this would help the reader to understand the availability of these 3 variables in the study population, and also assess whether the statistically insignificantly results could be a type 2 error.
Discussion
Line 281-283: Recommend the author to cite references for these breeds being more easily to have RHE with normal cardiac size and function.
Line 315: The author stated dogs with RHE have significantly different echocardiographic parameters than those without. This is expected if the author classifies the dog with and without RHE based on these 3 parameters.
I would, however, recommend the author include a more detailed description of the value of these 3 echocardiographic parameters in Table 3 to allow a more comprehensive understanding of the readers.
Author Response
Comment 1: Line 43: please list the reference for the aforementioned breeds being more frequently suggested RHE on radiography, especially for those non-brachycephalic breeds.
Response 1: Thank you for your insightful comment regarding the radiographic appearance of pseudo-RHE in specific breeds.
We appreciate your attention to detail. Upon careful consideration of your feedback, we acknowledge that we were unable to find precise literature supporting the observation of pseudo-RHE in the mentioned breeds. The statement was primarily based on the authors' clinical experience and observations.
We sincerely appreciate you bringing this to our attention. In light of your comment, we will revise the manuscript to clearly indicate that this observation is based on the authors' experience rather than published literature. We will ensure that the text accurately reflects the nature of this information.
In response to your comments, I have made the following additions to the manuscript:
“Based on the authors' experience, in certain breeds, such as brachycephalic dogs, Schnauzers, Dachshunds, Pomeranians, Yorkshire Terriers, and Miniature Pinschers, radiographic images frequently suggest apparent RHE (e.g., right heart bulging) despite normal cardiac size and function.”
Thank you again for your valuable input. Your feedback has certainly helped improve the accuracy and transparency of our manuscript.
Comment 2: Line70: I believe a more detailed description of the criteria of without signs of pulmonary hypertension should be described more detaily.
What do you mean: without signs of pulmonary hypertension?
For instance: are patients with trivial/physiological TR allowed to be included?, Are patients with trivial Pulmonary insufficiency allowed to be included, Are patients with mild TR but a normal estimated PAP allowed to be included?
I believe more detailed criteria would be beneficial for this manuscript.
Response 2: Thank you sincerely for your excellent observation and feedback. Your input has been invaluable, providing significant assistance and offering a detailed explanation of the criteria. I appreciate the opportunity to clarify my intended meaning.
I wanted to convey that we aimed to include cases with trivial or physiological tricuspid regurgitation in our study. In light of your comments, I have revised the relevant section as follows:
"Furthermore, individuals presenting with underlying conditions (e.g., coughing, dyspnea, signs of pulmonary hypertension, or right heart remodeling) that could potentially confound the experimental outcomes were excluded from the study cohort. However, patients with trivial or physiological tricuspid regurgitation were included, as these conditions were not indicative of pulmonary hypertension and were deemed unlikely to affect the study results."
This revision aims to provide a clearer and more accurate description of our inclusion and exclusion criteria, addressing the point you raised. Once again, I am grateful for your astute feedback, which has significantly enhanced the clarity and precision of our manuscript.
Comment 3: Line 71: The author cited the 2024 JVIM study for the right heart remodeling criteria, while this is a retrospective study between 2015 and 2024.
Does the author go back and review all the images and measure the right heart ratio based on the 2024 JVIM study?
If so, how many authors participate in the reviewing process?
I believe a more detailed description would benefit the reader to better understand the criteria.
Response 3: Thank you for your valuable feedback. Your comments have highlighted the need to reassess and clarify our criteria for right heart enlargement (RHE).
Regarding the citation of the 2024 JVIM study, I appreciate your bringing this to my attention. Allow me to explain our approach:
The criteria we used for RHE were based on RVEDA, RVWTd, and RAA indices. We initially cited the JVIM study because it utilized the same echocardiographic indices. However, I understand that this may have caused confusion rather than clarity.
In light of your feedback, I propose the following action:
We will remove the reference to the 2024 JVIM study to avoid any potential misunderstanding.
Instead, we will cite the original papers that established the criteria for each of these indices. Specifically:
- Gentile-Solomon, J.M.; Abbott, J.A. Conventional Echocardiographic Assessment of the Canine Right Heart: Reference In-tervals and Repeatability. J. Vet. Cardiol. 2016, 18, 234–247. https://doi.org/10.1016/j.jvc.2016.04.005.
- Visser, L.C.; Scansen, B.A.; Schober, K.E.; Bonagura, J.D. Echocardiographic assessment of right ventricular systolic function in conscious healthy dogs: repeatability and reference intervals. J. Vet. Cardiol. 2015, 17, 83–96. https://doi.org/10.1016/j.jvc.2014.10.003.
- Vezzosi, T.; Domenech, O.; Costa, G.; Marchesotti, F.; Venco, L.; Zini, E.; Del Palacio, M.J.F.; Tognetti, R. Echocardiographic evaluation of the right ventricular dimension and systolic function in dogs with pulmonary hypertension. J. Vet. Intern. Med. 2018, 32, 1541–1548. https://doi.org/10.1111/jvim.15253.
- Vezzosi, T.; Domenech, O.; Iacona, M.; Marchesotti, F.; Zini, E.; Venco, L.; Tognetti, R. Echocardiographic evaluation of the right atrial area index in dogs with pulmonary hypertension. J. Vet. Intern. Med. 2018, 32, 42–47. https://doi.org/10.1111/jvim.14896.
- Visser, L.C.; Im, M.K.; Johnson, L.R.; Stern, J.A. Diagnostic value of right pulmonary artery distensibility index in dogs with pulmonary hypertension: comparison with doppler echocardiographic estimates of pulmonary arterial pressure. J. Vet. Intern. Med. 2016, 30, 543–552. https://doi.org/10.1111/jvim.13911.
- Feldhütter, E.K.; Domenech, O.; Vezzosi, T.; et al. Echocardiographic reference intervals for right ventricular indices. in-cluding 3-dimensional volume and 2-dimensional strain measurements in healthy dogs. J. Vet. Intern. Med. 2022, 36, 8-19. https://doi.org/10.1111/jvim.16311.
This approach should provide a clearer foundation for our RHE criteria and eliminate any ambiguity caused by referencing more recent studies that may have used these indices in a different context.
Regarding your inquiry about the authors involved in radiographic and echocardiographic parameter measurements, I sincerely apologize for the inadvertent omission of this crucial information during the revision process. Your astute observation has allowed us to rectify this oversight.
We have now added the following sentence to clarify the measurement process:
"All radiographic/echocardiographic measurements were obtained by a single radiologist (D.V.M) under the supervision of a senior radiologist with more than 10 years of radiology experience."
To provide further detail, I would like to add that all measurements were performed twice by the same author/veterinarian (Kawon Choi, DVM), and the average of these two measurements was recorded as the final value. Your meticulous review has once again impressed me, and I am grateful for your attention to detail. These additions will undoubtedly enhance the methodological transparency of our study.
I sincerely appreciate your attention to detail, which has allowed us to improve the accuracy and clarity of our manuscript. If you have any further suggestions or concerns, please don't hesitate to let me know.
Thank you again for your invaluable input in refining our research.
Comment 4: Line 116, I suggest changing the description to:
right ventricular hypertrophy and/or enlargement was defined as an RVWTd index > 0.39 and/or an RVEDA index > 1.4.
I also suggested the author should clarify how many veterinarians/Authors have participated in the measurement of radiological and echocardiographic parameters, respectively. Multiple observers/raters would affect the variability and replicability of the study, especially when inter-observer variability assessment was not performed.
Response 4: Thank you for your valuable feedback and keen observations. Your suggestions have significantly improved the clarity and precision of our manuscript. I appreciate your recommendation for rephrasing, which has indeed resulted in a more polished expression.
As per your suggestion, we have revised the sentence to:
"Right ventricular hypertrophy and/or enlargement was defined as an RVWTd index > 0.39 and/or an RVEDA index > 1.4, while right atrial enlargement was defined as an RAAs index > 0.76."
Regarding your inquiry about the authors involved in radiographic and echocardiographic parameter measurements, I sincerely apologize for the inadvertent omission of this crucial information during the revision process. Your astute observation has allowed us to rectify this oversight.
We have now added the following sentence to clarify the measurement process:
"All radiographic/echocardiographic measurements were obtained by a single radiologist (D.V.M) under the supervision of a senior radiologist with more than 10 years of radiology experience."
To provide further detail, I would like to add that all measurements were performed twice by the same author/veterinarian (Kawon Choi, DVM), and the average of these two measurements was recorded as the final value. Your meticulous review has once again impressed me, and I am grateful for your attention to detail. These additions will undoubtedly enhance the methodological transparency of our study.
Thank you again for your invaluable input and for helping us improve the quality of our manuscript.
Comment 5: Line 183 I’m confused about the criteria of RHE and non-RHE groups of the study. In Line 71, the authors cite the 2024 JVIM study as the reference for right heart remodeling, while in Line 116, the authors state the criteria for RHE is based on RVEDA index, RVWTd index, and RAA index. But here the authors state the RAA, RVWTd, and RVEDA could be excluded if image quality was impossible to measure.
So my question is, what exactly are the criteria for the non-RHE group, is it based on the absence of TR and normal right heart ratio, or, based on RVEDA, RVWTd, and RAA.
A detailed description of the criteria of the non-RHE group is needed.
Response 5: I sincerely apologize for the repeated clarifications. Thank you for your patience and thorough review.
As mentioned in Line 183, we excluded measurements when image quality was insufficient. To clarify:
- For the RHE group: All echocardiographic parameters (RVEDA, RVWTd, RAA index) were measured for cardiac enlargement diagnosis.
- For the non-RHE group:
- We included only animals without relevant symptoms (cough, syncope, dyspnea, or other pulmonary hypertension symptoms).
- We checked for the presence of TR and measured echocardiographic parameters.
- At least one of the three echocardiographic parameters was measured for each animal and included only if within normal range.
- As you correctly pointed out, only animals with no significant TR (excluding trivial/physiological TR) and normal echocardiographic parameters were included in the non-RHE group.
We acknowledge that our current wording may cause confusion regarding whether we excluded animals with poor image quality or just omitted specific measurements. We will revise the manuscript to clarify this point.
We have added the following sentence to the manuscript:
"Echocardiographic parameters, including RAA, RVWTd, and RVEDA, were measured for RHE diagnosis."
If you feel any further modifications are necessary, please let us know, and we will be happy to make additional revisions to ensure clarity.
Thank you again for your valuable input and for helping us improve the quality and precision of our manuscript.
Comment 6: Line 197-198, the author stated there were no significant differences in RAA, RVWTd, and RVEDA when comparing the NBC group to other breeds. I suggested a table with the mean +/- SD of these three parameters would be beneficial for the reader to understand if it is still possible to have a difference in echocardiographic parameters between groups, despite not being statistically significant.
Line 197-198, I also recommend the table should include the number of dogs that could be successfully obtain RAA, RVWTd, and RVEDA values of each group, as this would help the reader to understand the availability of these 3 variables in the study population, and also assess whether the statistically insignificantly results could be a type 2 error.
Response 6: Thank you for your valuable feedback. I sincerely appreciate your suggestion, which will undoubtedly enhance the clarity and comprehensiveness of our manuscript.
In response to your comments, I will make the following additions to the table:
- The mean and standard deviation values for each echocardiographic index (RAA, RVWTd, RVEDA) for all groups.
- The p-values from the comparison between each group and the NBC (Non-Brachycephalic) group.
- The number of subjects for each index in each group.
I have added the following sentence to the main text to introduce this information:
"The mean, SD values, for the echocardiographic indices (RAA, RVWTd, RVEDA) of each group are summarized in Table 3."
Additionally, I will update Table 3 to include all of this information in a clear and organized manner. These additions will provide a more comprehensive view of our data and facilitate easier comparison between groups.

Thank you once again for your insightful suggestion, which has significantly improved the quality of our manuscript.
Comment 7: Line 281-283: Recommend the author to cite references for these breeds being more easily to have RHE with normal cardiac size and function.
Response 7: Thank you for your insightful comment regarding the radiographic appearance of pseudo-RHE in specific breeds.
We appreciate your attention to detail. Upon careful consideration of your feedback, we acknowledge that we were unable to find precise literature supporting the observation of pseudo-RHE in the mentioned breeds. The statement was primarily based on the authors' clinical experience and observations.
We sincerely appreciate you bringing this to our attention. In light of your comment, we will revise the manuscript to clearly indicate that this observation is based on the authors' experience rather than published literature. We will ensure that the text accurately reflects the nature of this information.
In response to your comments, I have made the following additions to the manuscript:
“Based on the authors' experience , certain breeds, including brachycephalic breeds, Schnauzers, Dachshunds, Pomeranians, Yorkshire Terriers, and Miniature Pinschers, frequently display radiographic signs of apparent RHE despite having normal cardiac size and function.”
Thank you again for your valuable input. Your feedback has certainly helped improve the accuracy and transparency of our manuscript.
Comment 8: Line 315: The author stated dogs with RHE have significantly different echocardiographic parameters than those without. This is expected if the author classifies the dog with and without RHE based on these 3 parameters.
I would, however, recommend the author include a more detailed description of the value of these 3 echocardiographic parameters in Table 3 to allow a more comprehensive understanding of the readers.
Response 8:
Thank you for your insightful comments. I greatly appreciate your guidance, which has significantly enhanced the clarity and comprehensiveness of our manuscript.
You are absolutely correct in pointing out that the significant differences between the RHE and non-RHE groups are an expected outcome, given that we used echocardiographic parameters to define these groups. This is an important distinction that readers should be aware of.
In response to your suggestion, we have added more detailed information to Table 4, including:
- Mean values
- Standard deviations
- Number of subjects for each parameter
These additions will provide readers with a more comprehensive understanding of our data and allow for a more nuanced interpretation of our results.
We believe that this enhanced presentation of data will enable readers to gain a more thorough insight into our study's findings and methodology. This level of detail should facilitate a better appreciation of the differences between the RHE and non-RHE groups, as well as the overall distribution of echocardiographic parameters across our study population.

Thank you once again for your valuable feedback. Your attention to detail has undoubtedly improved the quality and transparency of our research presentation. If you have any further questions or suggestions, please don't hesitate to let me know.
Round 2
Reviewer 1 Report
Comments and Suggestions for Authors
The authors have addressed all my concerns and went above and beyond to polish their work. I have no further requests.
Author Response
Comment: The authors have addressed all my concerns and went above and beyond to polish their work. I have no further requests.
Reviewer 2 Report
Comments and Suggestions for Authors
I appreciate the authors’ effort in addressing my questions and their commitment to refining the manuscript. Most of my concerns have been addressed; however, I still have several major issues that need to be resolved.
1. Concerns Regarding the Reference to the 2024 JVIM Study
In response to my previous comment, the authors stated:
“The criteria we used for RHE were based on RVEDA, RVWTd, and RAA indices. We initially cited the JVIM study because it utilized the same echocardiographic indices. However, I understand that this may have caused confusion rather than clarity. In light of your feedback, I propose the following action: We will remove the reference to the 2024 JVIM study to avoid any potential misunderstanding. Instead, we will cite the original papers that established the criteria for each of these indices.”
I appreciate the authors’ willingness to clarify the citation. However, I must point out that the JVIM study did not utilize these echocardiographic indices. The study focused on two-dimensional parameters and did not include RVEDA, RVWTd, or RAA indices, nor did it use these parameters as criteria for its study population. The authors should ensure that references are accurately cited to avoid misrepresentation.
2. Inclusion of Proper Units for Echocardiographic Parameters
I strongly recommend that the authors carefully review the manuscript and ensure that all echocardiographic parameters are accompanied by appropriate units wherever they appear.
3. Major Errors in the Manuscript
I have identified multiple fundamental errors that significantly impact the accuracy of the study:
• Lack of Proper References for Cut-Off Values
The manuscript defines right ventricular hypertrophy and/or enlargement as an RVWTd index > 0.39 and/or an RVEDA index > 1.4, and right atrial enlargement as an RAA index > 0.76. However, I was unable to find supporting references for these cut-off values in the cited literature. The authors should provide clear justification for these thresholds, supported by established references.
• Error in Table 3
The p-value currently appearing in the “NBC” row should likely be moved to the “Brachycephalic Breed” row.
• Inconsistent Indexing and Labeling in Tables 3 and 4
The authors must ensure that all listed echocardiographic parameters in Tables 3 and 4 are indexed. Additionally, the table descriptions should clearly indicate that these values are indices, and all units should be correctly included.
• Contradiction in Echocardiographic Parameter Values
A major issue exists in the reported echocardiographic parameters. If the authors’ criteria for differentiating RHE and non-RHE groups are applied, the echocardiographic parameter values for the non-RHE group already exceed the defined cut-off values. This is logically inconsistent and should not be possible. The authors must thoroughly review their dataset and confirm the accuracy of their reported values.
Recommendation
Despite the authors’ efforts to refine the manuscript, the presence of multiple fundamental errors significantly undermines the accuracy and validity of the study. Given these issues, I recommend rejecting the current manuscript in its present form. Thorough revisions are required before the study can be considered for publication.
Author Response
First and foremost, I would like to express my sincere apologies for not providing a response that adequately addressed your excellent initial feedback. I deeply regret the frequent errors that occurred and take full responsibility for them. Moving forward, I assure you that I will be more meticulous in checking my responses to minimize such mistakes and provide more thorough answers.
Furthermore, I am truly grateful that, despite these shortcomings, you have once again taken the time to provide us with high-quality feedback. I promise to respond with the utmost care and attention to detail, ensuring that this response surpasses the quality of my initial one. Your continued patience and guidance are immensely appreciated.
Comment 1: Concerns Regarding the Reference to the 2024 JVIM Study
In response to my previous comment, the authors stated:
“The criteria we used for RHE were based on RVEDA, RVWTd, and RAA indices. We initially cited the JVIM study because it utilized the same echocardiographic indices. However, I understand that this may have caused confusion rather than clarity. In light of your feedback, I propose the following action: We will remove the reference to the 2024 JVIM study to avoid any potential misunderstanding. Instead, we will cite the original papers that established the criteria for each of these indices.”
I appreciate the authors’ willingness to clarify the citation. However, I must point out that the JVIM study did not utilize these echocardiographic indices. The study focused on two-dimensional parameters and did not include RVEDA, RVWTd, or RAA indices, nor did it use these parameters as criteria for its study population. The authors should ensure that references are accurately cited to avoid misrepresentation.
Response 1:
Thank you for your valuable feedback. I apologize for the misrepresentation caused by my incorrect citation. I will remove the reference you mentioned and replace it with more appropriate citations. Your input is greatly appreciated.
The revised sentence is as follows:
However, patients with trivial or physiological tricuspid regurgitation were included, as these conditions were not indicative of pulmonary hypertension and were deemed unlikely to affect the study results. [5].
- Reinero, C.; Visser, L.C.; Kellihan, H.B.; Masseau, I.; Rozanski, E.; Clercx, C.; Williams, K.; Abbott, J.; Borgarelli, M.; Scansen, B.A. ACVIM consensus statement guidelines for the diagnosis, classification, treatment, and monitoring of pulmonary hypertension in dogs. J. Vet. Intern. Med. 2020, 34, 549-573. https://doi.org/10.1111/jvim.15725
Comment 2: Inclusion of Proper Units for Echocardiographic Parameters
I strongly recommend that the authors carefully review the manuscript and ensure that all echocardiographic parameters are accompanied by appropriate units wherever they appear.
Response 2:
Thank you for your valuable feedback. We have thoroughly reviewed the entire manuscript once again.
We have made the following revisions:
- (revisioned version script_line 126) Right ventricular hypertrophy and/or enlargement was defined as an RVWTd index (cm/kg0.250) > 0.39 and/or an RVEDA index (cm2/kg0.624) > 1.4, while right atrial enlargement was defined as an RAAs index (cm2/kg0.714) > 0.76
- (revisioned version script_Table3)

- (revisioned version script_Table4)

Comment 3-1:
- Lack of Proper References for Cut-Off Values
The manuscript defines right ventricular hypertrophy and/or enlargement as an RVWTd index > 0.39 and/or an RVEDA index > 1.4, and right atrial enlargement as an RAA index > 0.76. However, I was unable to find supporting references for these cut-off values in the cited literature. The authors should provide clear justification for these thresholds, supported by established references.
Response 3-1:
Thank you for bringing this matter to our attention. We appreciate your thorough feedback.
Regarding the source of the mentioned section, our paper draws significantly from the 2024 VRHi study (The vertebral right heart index: A new radiographic method to assess right heart enlargement in dogs. Vet. Radiol. Ultrasound 2024, 65, 596-602). We used the cut-off values from this study, and we will add this paper to our citations.
We also acknowledge a citation error in our manuscript (incorrectly citing reference 14 instead of 16). We have corrected this to the appropriate citation (16. Feldhütter, E.K.; Domenech, O.; Vezzosi, T.; et al. Echocardiographic reference intervals for right ventricular indices, including 3-dimensional volume and 2-dimensional strain measurements in healthy dogs. J. Vet. Intern. Med. 2022, 36, 8-19. https://doi.org/10.1111/jvim.16331).
We sincerely apologize for the improper citation and will rectify this by inserting the correct reference. Furthermore, we have thoroughly reviewed our citations to ensure accuracy throughout the manuscript.
The revised version (line 126) now reads:
"Right ventricular hypertrophy and/or enlargement was defined as an RVWTd index (cm/kg0.250) > 0.39 and/or an RVEDA index (cm2/kg0.624) > 1.4, while right atrial enlargement was defined as an RAAs index (cm2/kg0.714) > 0.76 [3,11,16]."
We have included the following references:
3. Puccinelli, C.; Vezzosi, T.; Grosso, G.; Citi, S.; Della Santa, D.; Buralli, C.; Marchetti, V.; Tognetti, R. The vertebral right heart index: A new radiographic method to assess right heart enlargement in dogs. Vet. Radiol. Ultrasound 2024, 65, 596-602. https//doi.org/10.1111/vru.13402.
Relevant quotes from this reference:
- "Right ventricular enlargement was defined using two-dimensional echocardiography as follows: right ventricular free wall thickness normalized for body weight >0.39cm/kg0.250 and/or right ventricular end-diastolic area normalized for body weight >1.4cm2/kg0.665. [11,16]"
- "Right atrial enlargement was defined as follows: right atrial end-systolic area normalized for body weight > 0.76 cm2/kg0.71.[11]"
11. Gentile-Solomon, J.M.; Abbott, J.A. Conventional Echocardiographic Assessment of the Canine Right Heart: Reference Intervals and Repeatability. J. Vet. Cardiol. 2016, 18, 234–247. https://doi.org/10.1016/j.jvc.2016.05.002.
16. Feldhütter, E.K.; Domenech, O.; Vezzosi, T.; Bussadori, C.; Zini, E.; Marchesotti, F.; Venco, L.; Tognetti, R. Echocardiographic reference intervals for right ventricular indices, including 3-dimensional volume and 2-dimensional strain measurements in healthy dogs. J. Vet. Intern. Med. 2022, 36, 8-19. https://doi.org/10.1111/jvim.16331.
We apologize for any confusion this may have caused and thank you for your careful review of our manuscript.
Comment 3-2:
- Error in Table 3
The p-value currently appearing in the “NBC” row should likely be moved to the “Brachycephalic Breed” row.
Response 3-2:
I sincerely apologize for the error and thank you for bringing it to our attention. We have made the necessary corrections as per your feedback. Moving forward, we will be even more meticulous in implementing revisions. We greatly appreciate your keen observation.
The revised table is attached below for your reference.
Thank you again for your time and valuable input. We are committed to maintaining the highest standards of accuracy and reliability in our research.
Comment 3-3:
- Inconsistent Indexing and Labeling in Tables 3 and 4
The authors must ensure that all listed echocardiographic parameters in Tables 3 and 4 are indexed. Additionally, the table descriptions should clearly indicate that these values are indices, and all units should be correctly included.
Response 3-3:
Thank you for your valuable feedback regarding the importance of clearly indicating the index nature of these measurements. We greatly appreciate your attention to detail.
We have revised the table to incorporate your suggestions, ensuring that all relevant values are clearly labeled as indices. Additionally, we have inserted the specific values as requested.
These modifications should provide a more precise and comprehensive representation of the data. We believe these changes will enhance the clarity and accuracy of our presentation.
Thank you once again for your insightful comments. Your input has significantly improved the quality of our work.
The updated table is attached below for your review.
- (revisioned version script_Table3)

- (revisioned version script_Table4)

Comment 3-4:
- Contradiction in Echocardiographic Parameter Values
A major issue exists in the reported echocardiographic parameters. If the authors’ criteria for differentiating RHE and non-RHE groups are applied, the echocardiographic parameter values for the non-RHE group already exceed the defined cut-off values. This is logically inconsistent and should not be possible. The authors must thoroughly review their dataset and confirm the accuracy of their reported values.
Response 3-4:
Thank you for your valuable feedback. I sincerely apologize for not adequately addressing this point in the previous revision. Your insight is greatly appreciated.
We acknowledge that comparing the RHE and non-RHE groups using p-values is not appropriate, given that these groups were initially divided based on echocardiographic parameters. As a result, we have decided to remove these comparisons from both the main text and the table.
We are grateful for your valuable insight. The revised sections are as follows:
- (Revised version script, line 151) "Additionally, t-tests were used to compare differences in RL/VD VRHi and echocardiographic parameters between dogs with and without RHE in NBC and brachycephalic groups."
- (Revised version script, Table 4) The p-value column for echocardiographic parameters has been removed.
- (Revised version script, line 257) In the brachycephalic and NBC groups, individuals with RHE exhibited significantly higher VRHi and echocardiographic indices compared to those without RHE (all P < 0.001)
- (Revised version script, line 340) Comparative analysis of dogs with and without RHE revealed statistically significant differences in RL and VD VRHi and echocardiographic parameters (RAAs, RVWTd, RVEDA index) across brachycephalic and NBC cohorts.
We believe these changes address the concerns you raised and improve the overall quality and accuracy of our manuscript. Thank you again for your thorough review and insightful comments. Your feedback has been instrumental in enhancing our work.
Round 3
Reviewer 2 Report
Comments and Suggestions for Authors
I thank the author's endeavor and hard work they put in to answer my previous questio and comment. I do believe the manuscript has been significantly improved.
Most of my previous questions and comments have been answered.
However, the value of echocardiographic parameters in Table 4 remained logically inconsistent. (I also was concerned about the reported value in Table 3)
To be specific, I quote the author's echocardiographic criteria to differentiate RHE and non-RHE
Right ventricular hypertrophy and/or enlargement was defined as an RVWTd index (cm/kg0.250) > 0.39 and/or an RVEDA index (cm2 /kg0.624) > 1.4, while right atrial enlargement was defined as an RAAs index (cm2 /kg0.714) > 0.76 [3,11,16].
Therefore, using a RVWTd index > 0.39, RVEDA > 1.4, and RAAs > 0.76
How does the non-RHE group of NBC have a mean RVWTd index of 3.01, an RVEDA index of 59.34, and an RAA index of 43.34?
How does the non-RHE brachycephalic group have a mean RVWTd index of 3.13, RVEDA index of 53.91, and RAA index of 43.01?
Again, this is logically inconsistent and should not be possible.
Author Response
Comment 1:
However, the value of echocardiographic parameters in Table 4 remained logically inconsistent. (I also was concerned about the reported value in Table 3)
To be specific, I quote the author's echocardiographic criteria to differentiate RHE and non-RHE
Right ventricular hypertrophy and/or enlargement was defined as an RVWTd index (cm/kg0.250) > 0.39 and/or an RVEDA index (cm2 /kg0.624) > 1.4, while right atrial enlargement was defined as an RAAs index (cm2 /kg0.714) > 0.76 [3,11,16].
Therefore, using a RVWTd index > 0.39, RVEDA > 1.4, and RAAs > 0.76
How does the non-RHE group of NBC have a mean RVWTd index of 3.01, an RVEDA index of 59.34, and an RAA index of 43.34?
How does the non-RHE brachycephalic group have a mean RVWTd index of 3.13, RVEDA index of 53.91, and RAA index of 43.01?
Again, this is logically inconsistent and should not be possible.
Response 1:
Dear Reviewer,
First, I sincerely apologize for causing you to point out numerous issues due to my frequent mistakes. As mentioned, it seems that the contradictory errors occurred due to confusion between mm and cm.
I would like to express my apologies once again and inform you that I have corrected all the related tables.

Although I have been expressing my gratitude since the first revision, I truly appreciate your meticulous review and thorough examination. As this is my first paper, there were many areas that needed improvement. I am deeply grateful for your guidance in supplementing both academic and technical aspects, and for your suggestions on how to enhance the paper.
Thank you for your valuable input and support.
Round 4
Reviewer 2 Report
Comments and Suggestions for Authors
I appreciate the author's hard work in improving the manuscript and answering my concerns and questions.
I believe the manuscript has been significantly improved and is suitable to be published in its current form.